# LAMBDANET: PROBABILISTIC TYPE INFERENCE USING GRAPH NEURAL NETWORKS

**Jiayi Wei, Maruth Goyal, Greg Durrett, Isil Dillig**
Department of Computer Science
University of Texas at Austin
`{jiayi, maruth, gdurrett, isil}@cs.utexas.edu`

## ABSTRACT

As *gradual typing* becomes increasingly popular in languages like Python and TypeScript, there is a growing need to infer type annotations automatically. While type annotations help with tasks like code completion and static error catching, these annotations cannot be fully determined by compilers and are tedious to annotate by hand. This paper proposes a probabilistic type inference scheme for TypeScript based on a graph neural network. Our approach first uses lightweight source code analysis to generate a program abstraction called a *type dependency graph*, which links type variables with logical constraints as well as name and usage information. Given this program abstraction, we then use a graph neural network to propagate information between related type variables and eventually make type predictions. Our neural architecture can predict both standard types, like `number` or `string`, as well as user-defined types that have not been encountered during training. Our experimental results show that our approach outperforms prior work in this space by $14\%$ (absolute) on library types, while having the ability to make type predictions that are out of scope for existing techniques.

## 1 INTRODUCTION

Dynamically typed languages like Python, Ruby, and Javascript have gained enormous popularity over the last decade, yet their lack of a static type system comes with certain disadvantages in terms of maintainability (Hanenberg et al., 2013), the ability to catch errors at compile time, and code completion support (Gao et al., 2017). *Gradual typing* can address these shortcomings: program variables have *optional* type annotations so that the type system can perform static type checking whenever possible (Siek & Taha, 2007; Chung et al., 2018). Support for gradual typing now exists in many popular programming languages (Bierman et al., 2014; Vitousek et al., 2014), but due to their heavy use of dynamic language constructs and the absence of principal types (Ancona & Zucca, 2004), compilers cannot perform type inference using standard algorithms from the programming languages community (Bierman et al., 2014; Traytel et al., 2011; Pierce & Turner, 2000), and manually adding type annotations to existing codebases is a tedious and error-prone task. As a result, legacy programs in these languages do not reap all the benefits of gradual typing.

To reduce the human effort involved in transitioning from untyped to statically typed code, this work focuses on a learning-based approach to automatically inferring likely type annotations for untyped (or partially typed) codebases. Specifically, we target TypeScript, a gradually-typed variant of Javascript for which plenty of training data is available in terms of type-annotated programs. While there has been some prior work on inferring type annotations for TypeScript using machine learning (Hellendoorn et al., 2018; Raychev et al., 2015), prior work in this space has several shortcomings. First, inference is restricted to a finite dictionary of types that have been observed during training time—i.e., they cannot predict any user-defined data types. Second, even without considering user-defined types, the accuracy of these systems is relatively low, with the current state-of-the-art achieving 56.9% accuracy for primitive/library types (Hellendoorn et al., 2018). Finally, these techniques can produce inconsistent results in that they may predict different types for different token-level occurrences of the same variable.

```
1    class MyNetwork {
2        name: string;  time: number;
3        forward(x: Tensor, y: Tensor): Tensor {
4            return x.concat(y) * 2;
5        }
6    }
7    // more classes ...
8    function restore (network: MyNetwork): void {
9        network.time = readNumber("time.txt");
10       // more code...
11   }
```

Figure 1: A motivating example: Given an unannotated version of this TypeScript program, a traditional rule-based type inference algorithm cannot soundly deduce the true type annotations (shown in green).

In this paper, we propose a new probabilistic type inference algorithm for TypeScript to address these shortcomings using a graph neural network architecture (GNN) (Veličković et al., 2018; Li et al., 2016; Mou et al., 2016). Our method uses lightweight source code analysis to transform the program into a new representation called a *type dependency graph*, where nodes represent type variables and labeled hyperedges encode relationships between them. In addition to expressing logical constraints (e.g., subtyping relations) as in traditional type inference, a type dependency graph also incorporates contextual hints involving naming and variable usage.

Given such a type dependency graph, our approach uses a GNN to compute a vector embedding for each type variable and then performs type prediction using a pointer-network-like architecture (Vinyals et al., 2015). The graph neural network itself requires handling a variety of hyperedge types—some with variable numbers of arguments—for which we define appropriate graph propagation operators. Our prediction layer compares the vector embedding of a type variable with vector representations of candidate types, allowing us to flexibly handle user-defined types that have not been observed during training. Moreover, our model predicts consistent type assignments by construction because it makes variable-level rather than token-level predictions.

We implemented our new architecture as a tool called LAMBDANET and evaluated its performance on real-world TypeScript projects from Github. When only predicting library types, LAMBDANET has a top1 accuracy of $75.6\%$, achieving a significant improvement over DeepTyper ($61.5\%$). In terms of overall accuracy (including user-defined types), LAMBDANET achieves a top1 accuracy of around $64.2\%$, which is $55.2\%$ (absolute) higher than the TypeScript compiler.

**Contributions.** This paper makes the following contributions: (1) We propose a probabilistic type inference algorithm for TypeScript that uses deep learning to make predictions from the type dependency graph representation of the program. (2) We describe a technique for computing vector embeddings of type variables using GNNs and propose a pointer-network-like method to predict user-defined types. (3) We experimentally evaluate our approach on hundreds of real-world TypeScript projects and show that our method significantly improves upon prior work.

## 2 MOTIVATING EXAMPLE AND PROBLEM SETTING

Figure 1 shows a (type-annotated) TypeScript program. Our goal in this work is to infer the types shown in the figure, given an *unannotated* version of this code. We now justify various aspects of our solution using this example.

**Typing constraints.** The use of certain functions/operators in Figure 1 imposes hard constraints on the types that can be assigned to program variables. For example, in the `forward` function, variables `x`, `y` must be assigned a type that supports a `concat` operation; hence, `x`, `y` could have types like `string`, `array`, or `Tensor`, but not, for example, `boolean`. This observation motivates us to incorporate typing constraints into our model.

**Contextual hints.** Typing constraints are not always sufficient for determining the intended type of a variable. For example, for variable `network` in function `restore`, the typing constraints require `network`'s type to be a class with a field called `time`, but there can be *many* classes that have such an attribute (e.g., `Date`). However, the similarity between the variable name `network`

```
1      var c1: τ₈ = class MyNetwork {
2          name: τ₁;   time: τ₂;
3          var m1: τ₉ = function forward(x: τ₃, y: τ₄):τ₅ {
4              var v1: τ₁₀ = x.concat;   var v2: τ₁₁ = v1(y);
5              var v3: τ₁₂ = v2.TIMES_OP; var v4: τ₁₃ = v3(NUMBER);
6              return v4;
7          }
8      }  // more classes...
9      var f1:τ₁₄ = function restore (network: τ₆): τ₇ {
10         var v3: τ₁₅ = network.time;
11         var v4: τ₁₆ = readNumber(STRING);
12         network.time = v4;  // more code...
13     }
```

Figure 2: An intermediate representation of the (unannotated version) program from Figure 1. The $\tau_i$ represent type variables, among which $\tau_8$–$\tau_{16}$ are newly introduced for intermediate expressions.

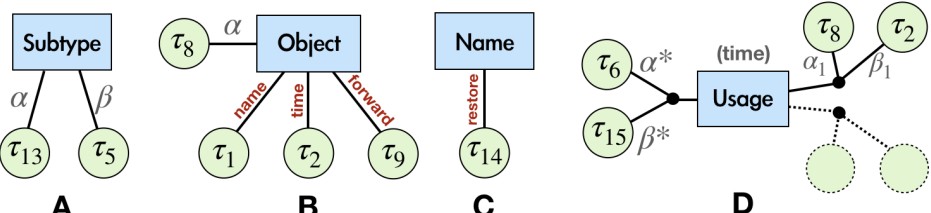

Figure 3: Example hyperedges for Figure 2. Edge labels in gray (resp. red) are positional arguments (resp. identifiers). **(A)** The return statement at line 6 induces a subtype relationship between $\tau_{13}$ and $\tau_5$. **(B)** MyNetwork $\tau_8$ declares attributes name $\tau_1$ and time $\tau_2$ and method forward $\tau_9$. **(C)** $\tau_{14}$ is associated with a variable whose named is restore. **(D)** Usage hyperedge for line 10 connects $\tau_6$ and $\tau_{15}$ to all classes with a time attribute.

and the class name MyNetwork hints that network might have type MyNetwork. Based on this belief, we can further propagate the return type of the library function readNumber (assuming we know it is number) to infer that the type of the time field in MyNetwork is likely to be number.

**Need for type dependency graph.** There are many ways to view programs—e.g., as token sequences, abstract syntax trees, control flow graphs, etc. However, none of these representations is particularly helpful for inferring the most likely type annotations. Thus, our method uses static analysis to infer a set of *predicates* that are relevant to the type inference problem and represents these predicates using a program abstraction called the *type dependency graph*.

**Handling user-defined types.** As mentioned in Section 1, prior techniques can only predict types seen during training. However, the code from Figure 1 defines its own class called MyNetwork and later uses a variables of type MyNetwork in the restore method. A successful model for this task therefore must dynamically make inferences about user-defined types based on their definitions.

## 2.1 PROBLEM SETTING

Our goal is to train a type inference model that can take as input an entirely (or partially) unannotated TypeScript project $g$ and output a probability distribution of types for each missing annotation. The prediction space is $\mathcal{Y}(g) = \mathcal{Y}_{\text{lib}} \cup \mathcal{Y}_{\text{user}}(g)$, where $\mathcal{Y}_{\text{user}}(g)$ is the set of all user-defined types (classes/interfaces) declared within $g$, and $\mathcal{Y}_{\text{lib}}$ is a fixed set of commonly-used library types.

Following prior work in this space (Hellendoorn et al., 2018; Raychev et al., 2015; Xu et al., 2016), we limit the scope of our prediction to non-polymorphic and non-function types. That is, we do not distinguish between types such as List<T>, List<number>, List<string> etc., and consider them all to be of type List. Similarly, we also collapse function types like number → string and string → string into a single type called Function. We leave the extension of predicting structured types as future work.

Table 1: Different types of hyperedges used in a type dependency graph.

| Type | Edge | Description |
|---|---|---|
| *Logical* | | |
| FIXED | $\text{Bool}(\alpha)$ | $\alpha$ is used as boolean |
| FIXED | $\text{Subtype}(\alpha, \beta)$ | $\alpha$ is a subtype of $\beta$ |
| FIXED | $\text{Assign}(\alpha, \beta)^\dagger$ | $\beta$ is assigned to $\alpha$ |
| NARY | $\text{Function}(\alpha, \beta_1, \ldots, \beta_k, \beta^*)$ | $\alpha = (\beta_1, \ldots, \beta_k) \rightarrow \beta^*$ |
| NARY | $\text{Call}(\alpha, \beta^*, \beta_1, \ldots, \beta_k)$ | $\alpha = \beta^*(\beta_1, \ldots, \beta_k)$ |
| NARY | $\text{Object}_{l_1, \ldots, l_k}(\alpha, \beta_1, \ldots, \beta_k)$ | $\alpha = \{l_1 : \beta_1, \ldots, l_k : \beta_k\}$ |
| FIXED | $\text{Access}_l(\alpha, \beta)$ | $\alpha = \beta.l$ |
| *Contextual* | | |
| FIXED | $\text{Name}_l(\alpha)$ | $\alpha$ has name $l$ |
| FIXED | $\text{NameSimilar}(\alpha, \beta)$ | $\alpha, \beta$ have similar names |
| NPAIRS | $\text{Usage}_l((\alpha^*, \beta^*), (\alpha_1, \beta_1), \ldots, (\alpha_k, \beta_k))$ | usages involving name $l$ |

$\dagger$ Although assignment is a special case of a subtype constraint, we differentiate them because these edges appear in different contexts and having uncoupled parameters for these two edge types is beneficial.

# 3   TYPE DEPENDENCY GRAPH

A type dependency graph $\mathcal{G} = (N, E)$ is a hypergraph where nodes $N$ represent type variables and labeled hyperedges $E$ encode relationships between them. We extract the type dependency graph of a given TypeScript program by performing static analysis on an *intermediate representation* of its source code, which allows us to associate a unique variable with each program sub-expression. As an illustration, Figure 2 shows the intermediate representation of the code from Figure 1.

Intuitively, a type dependency graph encodes properties of type variables as well as relationships between them. Each hyperedge corresponds to one of the predicates shown in Table 1. We partition our predicates (i.e., hyperedges) into two classes, namely *Logical* and *Contextual*, where the former category can be viewed as imposing hard constraints on type variables and the latter category encodes useful hints extracted from names of variables, functions, and classes.

Figure 3 shows some of the hyperedges in the type dependency graph $\mathcal{G}$ extracted from the intermediate representation in Figure 2. As shown in Figure 3(A), our analysis extracts a predicate $\text{Subtype}(\tau_{13}, \tau_5)$ from this code because the type variable associated with the returned expression v4 must be a subtype of the enclosing function's return type. Similarly, as shown in Figure 3(B), our analysis extracts a predicate $\text{Object}_{\text{name,time,forward}}(\tau_8, \tau_1, \tau_2, \tau_9)$ because $\tau_8$ is an object type whose name, time, and forward members are associated with type variables $\tau_1, \tau_2, \tau_9$, respectively.

In contrast to the Subtype and Object predicates that impose hard constraints on type variables, the next two hyperedges shown in Figure 3 encode contextual clues obtained from variable names. Figure 3(C) indicates that type variable $\tau_{14}$ is associated with an expression named restore. While this kind of naming information is invisible to TypeScript's structural type system (**?**), it serves as a useful input feature for our GNN architecture described in Section 4.

In addition to storing the unique variable name associated with each type variable, the type dependency graph also encodes similarity between variable and class names. The names of many program variables mimic their types: for example, instances of a class called MyNetwork might often be called network or network1. To capture this correspondence, our type dependency graph also contains a hyperedge called NameSimilar that connects type variables $\alpha$ and $\beta$ if their corresponding tokenized names have a non-empty intersection.[1]

As shown in Table 1, there is a final type of hyperedge called Usage that facilitates type inference of object types. In particular, if there is an object access var y = x.l, we extract the predicate $\text{Usage}_l((\tau_x, \tau_y), (\alpha_1, \beta_1), \ldots, (\alpha_k, \beta_k))$ to connect x and y's type variables with *all* classes $\alpha_i$ that contain an attribute/method $\beta_i$ whose name is l. Figure 3 shows a Usage hyperedge extracted from the code in Figure 2. As we will see in the next section, our GNN architecture utilizes a special attention mechanism to pass information along these usage edges.

---

[1]During tokenization, we split identifier names into tokens based on underscores and camel case naming. More complex schemes are possible, but we found this simple method to be effective.

# 4    NEURAL ARCHITECTURE

Our neural architecture for making type predictions consists of two main parts. First, a graph neural network passes information along the type dependency graph to produce a vector-valued embedding for each type variable based on its neighbors. Second, a pointer network compares each variable's type embedding to the embedding vectors of candidate types (both computed from the previous phase) to place a distribution over possible type assignments.

Given a type dependency graph $\mathcal{G} = (N, E)$, we first to compute a vector embedding $\mathbf{v}_n$ for each $n \in N$ such that these vectors implicitly encode type information. Because our program abstraction is a graph, a natural choice is to use a graph neural network architecture. From a high level, this architecture takes in initial vectors $\mathbf{v}_n^0$ for each node $n$, performs $K$ rounds of message-passing in the graph neural network, and returns the final representation for each type variable.

In more detail, let $\mathbf{v}_n^t$ denote the vector representation of node $n$ at the $t$th step, where each round consists of a *message passing* and an *aggregation* step. The message passing step computes a vector-valued update to send to the $j$th argument of each hyper-edge $e \in E$ connecting nodes $p_1, \dots, p_a$. Then, once all the messages have been computed, the *aggregation* step computes a new embedding $\mathbf{v}_n^t$ for each $n$ by combining all messages sent to $n$:

$$\mathbf{m}_{e,p_j}^t = \mathrm{Msg}_{e,j}(\mathbf{v}_{p_1}^{t-1}, \dots, \mathbf{v}_{p_a}^{t-1}) \quad \mathbf{v}_n^t = \mathrm{Aggr}(\mathbf{v}_n^{t-1}, \{\mathbf{m}_{e,n}^t | e \in \mathcal{N}(n)\})$$

Here, $\mathcal{N}$ is the neighborhood function, and $\mathrm{Msg}_e$ denotes a particular neural operation that depends on the type of the edge (FIXED, NARY, or NPAIRS), which we will describe later.

**Initialization.**    In our GNN, nodes correspond to type variables and each type variable is associated either with a program variable or a constant. We refer to nodes representing constants (resp. variables) as *constant (resp. variable) nodes*, and our initialization procedure works differently depending on whether or not $n$ is a constant node. Since the types of each constant are known, we set the initial embedding for each constant node of type $\tau$ (e.g., `string`) to be a trainable vector $\mathbf{c}_\tau$ and do not update it during GNN iterations (i.e., $\forall t, \mathbf{v}_n^t = \mathbf{c}_\tau$). On the other hand, if $n$ is a variable node, then we have no information about its type during initialization; hence, we initialize all variable nodes using a generic trainable initial vector (i.e., they are initialized to the *same* vector but updated to different values during GNN iterations).

**Message passing.**    Our $\mathrm{Msg}$ operator depends on the category of edge it corresponds to (see Table 1); however, weights are shared between all instances of the same hyperedge type. In what follows, we describe the neural layer that is used to compute messages for each type of hyperedge:

- FIXED: Since these edges correspond to fixed arity predicates (and the position of each argument matters), we compute the message of the $j$th argument by first concatenating the embedding vector of all arguments and then feed the result vector to a 2-layer MLP for the $j$th argument. In addition, since hyperedges of type *Access* have an identifier, we also embed the identifier as a vector and treat it as an extra argument. (We describe the details of identifier embedding later in this section.)

- NARY: Since NARY edges connect a variable number of nodes, we need an architecture that can deal with this challenge. In our current implementation of LAMBDANET, we use a simple architecture that is amenable to batching. Specifically, given an NARY edge $E_{l_1,\dots,l_k}(\alpha, \beta_1, \dots, \beta_k)$ (for *Function* and *Call*, the labels $l_j$ are argument positions), the set of messages for $\alpha$ is computed as $\{\mathrm{MLP}_\alpha(\mathbf{v}_{l_j} \| \mathbf{v}_{\beta_j}) \mid j = 1 \dots k\}$, and the message for each $\beta_j$ is computed as $\mathrm{MLP}_\beta(\mathbf{v}_{l_j} \| \mathbf{v}_\alpha)$. Observe that we compute $k$ different messages for $\alpha$, and the message for each $\beta_j$ only depends on the vector embedding of $\alpha$ and its position $j$, but not the vector embeddings of other $\beta_j$'s.[2]

- NPAIRS: This is a special category associated with $\mathrm{Usage}_l((\alpha^*, \beta^*), (\alpha_1, \beta_1), \dots, (\alpha_k, \beta_k))$. Recall that this kind of edge arises from expressions of the form $b = a.l$ and is used to connect $a$ and $b$'s type variables with all classes $\alpha_i$ that contain an attribute/method $\beta_i$ with label $l$. Intuitively, if $a$'s type embedding is very similar to a type $C$, then $b$'s type will likely be the same as $C.l$'s type. Following this reasoning, we use dot-product based attention to compute the messages for $\alpha^*$ and $\beta^*$. Specifically, we use $\alpha^*$ and $\alpha_j$'s as attention keys and $\beta_j$'s as attention values to compute the

---

[2]In our current implementation, this is reducible to multiple FIXED edges. However, NARY edges could generally use more complex pooling over their arguments to send more sophisticated messages.

message for $\beta^*$ (and switch the key-value roles to compute the message for $\alpha^*$):

$$\mathbf{m}_{e,\beta^*}^t = \sum_j w_j \mathbf{v}_{\beta_j}^{t-1} \qquad \mathbf{w} = \mathrm{softmax}(\mathbf{a}) \qquad a_j = \mathbf{v}_{\alpha_j} \cdot \mathbf{v}_{\alpha^*}$$

**Aggregation.** Recall that the aggregation step combines all messages sent to node $n$ to compute the new embedding $v_n^t$. To achieve this goal, we use a variant of the attention-based aggregation operator proposed in graph attention networks (Veličković et al., 2018).

$$v_n^t = \mathrm{Aggr}(v_n^{t-1}, \{m_{e,n}^t | e \in \mathcal{N}(n)\}) = v_n^{t-1} + \sum_{e \in \mathcal{N}(n)} w_e \mathbf{M}_1 m_{e,n}^t \tag{1}$$

where $w_e$ is the attention weight for the message coming from edge $e$. Specifically, the weights $w_e$ are computed as $\mathrm{softmax}(\mathbf{a})$, where $a_e = \mathrm{LeakyReLu}(v_n^{t-1} \cdot \mathbf{M}_2 m_{e,n}^t)$, and $\mathbf{M}_1$ and $\mathbf{M}_2$ are trainable matrices. Similar to the original GAT architecture, we set the slope of the LeakyReLu to be $0.2$, but we use dot-product to compute the attention weights instead of a linear model.

**Identifier embedding.** Like in Allamanis et al. (2017), we break variable names into word tokens according to camel case and underscore rules and assign a trainable vector for all word tokens that appear more than once in the training set. For all other tokens, unlike Allamanis et al. (2017), which maps them all into one single `<Unknown>` token, we randomly mapped them into one of the `<Unknown-i>` tokens, where $i$ ranges from 0 to 50 in our current implementation. This mapping is randomly constructed every time we run the GNN and hence helps our neural networks to distinguish different tokens even if they are rare tokens. We train these identifier embeddings end-to-end along with the rest of our architecture.

**Prediction Layer.** For each type variable $n$ and each candidate type $c \in \mathcal{Y}(g)$, we use a MLP to compute a compatibility score $s_{n,c} = \mathrm{MLP}(\mathbf{v}_n, \mathbf{u}_c)$, where $\mathbf{u}_c$ is the embedding vector for $c$. If $c \in \mathcal{Y}_{\mathrm{lib}}$, $\mathbf{v}_c$ is a trainable vector for each library type $c$; if $c \in \mathcal{Y}_{\mathrm{user}}(g)$, then it corresponds to a node $n_c$ in the type dependency graph of $g$, so we just use the embedding vector for $n_c$ and set $\mathbf{u}_c = \mathbf{v}_{n_c}$. Formally, this approach looks like a pointer network (Vinyals et al., 2015), where we use the embeddings computed during the forward pass to predict "pointers" to those types.

Given these compatibility scores, we apply a softmax layer to turn them into a probability distribution. i.e., $P_n(c|g) = \exp(s_{n,c})/\sum_{c'} \exp(s_{n,c'})$. During test time, we max over the probabilities to compute the most likely (or top-N) type assignments.

## 5 EVALUATION

In this section, we describe the results of our experimental evaluation, which is designed to answer the following questions: (1) How does our approach compare to previous work? (2) How well can our model predict user-defined types? (3) How useful is each of our model's components?

**Dataset.** Similar to Hellendoorn et al. (2018), we train and evaluate our model on popular open-source TypeScript projects taken from Github. Specifically, we collect 300 popular TypeScript projects from Github that contain between 500 to 10,000 lines of code and where at least 10% of type annotations are user-defined types. Note that each project typically contains hundreds to thousands of type variables to predict, and these projects in total contain about 1.2 million lines of TypeScript code. Among these 300 projects, we use 60 for testing, 40 for validation, and the remainder for training.

**Code Duplication.** We ran jscpd[3] on our entire data set and found that only 2.7% of the code is duplicated. Furthermore, most of these duplicates are intra-project. Thus, we believe that code duplication is not a severe problem in our dataset.

**Preprocessing.** Because some of the projects in our benchmark suite are only sparsely type annotated, we augment our labeled training data by using the forward type inference functionality provided by the TypeScript compiler.[4] The compiler cannot infer the type of every variable and leaves many labeled as `any` during failed inference; thus, we exclude `any` labels in our data set.

---

[3] A popular code duplication detection tool, available at `https://github.com/kucherenko/jscpd`.

[4] Like in many modern programming languages with forward type inference (e.g., Scala, C#, Swift), a TypeScript programmer does not need to annotate every definition in order to fully specify the types of a

Furthermore, at test time, we evaluate our technique only on annotations that are manually added by developers. This is the same methodology used by Hellendoorn et al. (2018), and, since developers often add annotations where code is most unclear, this constitutes a challenging setting for type prediction.

**Prediction Space.**   As mentioned in Section 2.1, our approach takes an entire TypeScript project $g$ as its input, and the corresponding type prediction space is $\mathcal{Y}(g) = \mathcal{Y}_{\text{lib}} \cup \mathcal{Y}_{\text{user}}(g)$. In our experiments, we set $\mathcal{Y}_{\text{user}}(g)$ to be all classes/interfaces defined in $g$ (except when comparing with DeepTyper, where we set $\mathcal{Y}_{\text{user}}(g)$ to be empty), and for $\mathcal{Y}_{\text{lib}}$, we select the top-100 most common types in our training set. Note that this covers $98\%$ (resp. $97.5\%$) of the non-`any` annotations for the training (resp. test) set.

**Hyperparameters**   We selected hyperparameters by tuning on a validation set as we were developing our model. We use 32-dimensional type embedding vectors, and all MLP transformations in our model use one hidden layer of 32 units, except the MLP for computing scores in the prediction layer, which uses three hidden layers of sizes 32,16, and 8 (and size 1 for output). GNN message-passing layers from different time steps have independent weights.

We train our model using Adam (Kingma & Ba, 2014) with default parameters ($\alpha = 0.9$, $\beta = 0.999$) and set the learning rate to be $10^{-3}$ initially but linearly decrease it to $10^{-4}$ until the 30th epoch. We use a weight decay of $10^{-4}$ for regularization and stop the training once the loss on validation set starts to increase (which usually happens around 30 epochs). We use the type annotations from a single project as a minibatch and limit the maximal batch size (via downsampling) to be the median of our training set to prevent any single project from having too much influence.

**Implementation Details.**   We implemented LAMBDANET in Scala, building on top of the Java high-performance Tensor library *Nd4j*(nd4), and used a custom automatic differentiation library to implement our GNN. Our GNN implementation does not use an adjacency matrix to represent GNN layers; instead, we build the hyperedge connections directly from our type dependency graph and perform batching when computing the messages for all hyperedges of the same type.

**Code Repository.**   We have made our code publicly available on Github.[5]

## 5.1   COMPARISON WITH DEEPTYPER

In this experiment, we compare LAMBDANET's performance with DeepTyper (Hellendoorn et al., 2018), which treats programs as sequences of tokens and uses a bidirectional RNN to make type predictions. Since DeepTyper can only predict types from a fixed vocabulary, we fix both LAMBDANET and DeepTyper's prediction space to $\mathcal{Y}_{\text{lib}}$ and measure their corresponding top-1 accuracy.

The original DeepTyper model makes predictions for each variable occurrence rather than declaration. In order to conduct a meaningful comparison between DeepTyper and LAMBDANET, we implemented a variant of DeepTyper that makes a single prediction for each variable (by averaging over the RNN internal states of all occurrences of the same variable before making the prediction). Moreover, for a fair comparison, we made sure both DeepTyper and LAMBDANET are using the same improved naming feature that splits words into tokens.

Our main results are summarized below, where the Declaration (resp. Occurrence) column shows accuracy per variable declaration (resp. token-level occurrence). Note that we obtain occurrence-level accuracy from declaration-level accuracy by weighting each variable by its number of occurrences.

| Model | Top1 Accuracy (%) | |
| --- | --- | --- |
| | *Declaration* | *Occurrence* |
| DeepTyper | 61.5 | 67.4 |
| LAMBDANET$_{\text{lib}}$ (K=6) | 75.6 | 77.0 |

---

program. Instead, they only need to annotate some key places (e.g., function parameters and return types, class members) and let the forward inference algorithm to figure out the rest of the types. Therefore, in our training set, we can keep the user annotations on these key places and run the TS compiler to recover these implicitly specified types as additional labels.

[5]See `https://github.com/MrVPlusOne/LambdaNet`.

Table 2: Accuracy when predicting all types.

| Model | Top1 Accuracy (%) | | | Top5 Accuracy (%) | | |
|---|---|---|---|---|---|---|
| | $\mathcal{Y}_{\text{user}}$ | $\mathcal{Y}_{\text{lib}}$ | *Overall* | $\mathcal{Y}_{\text{user}}$ | $\mathcal{Y}_{\text{lib}}$ | *Overall* |
| TS COMPILER | 2.66 | 14.39 | 8.98 | - | - | - |
| SIMILARNAME | 24.1 | 0.78 | 15.7 | 42.5 | 3.19 | 28.4 |
| LAMBDANET (K=6) | 53.4 | 66.9 | 64.2 | 77.7 | 86.2 | 84.5 |

Table 3: Performance of different GNN iterations (left) and ablations (right).

| K | Top1 Accuracy (%) | | | Ablation (K = 4) | Top1 Accuracy (%) | | |
|---|---|---|---|---|---|---|---|
| | $\mathcal{Y}_{\text{user}}$ | $\mathcal{Y}_{\text{lib}}$ | *Overall* | | $\mathcal{Y}_{\text{user}}$ | $\mathcal{Y}_{\text{lib}}$ | *Overall* |
| 6 | 53.4 | 66.9 | 64.2 | LAMBDANET | 48.4 | 65.5 | 62.0 |
| 4 | 48.4 | 65.5 | 62.0 | No Attention in NPAIR | 44.1 | 57.6 | 54.9 |
| 2 | 47.3 | 61.7 | 58.8 | No *Contextual* | 27.2 | 52.6 | 47.5 |
| 1 | 16.8 | 48.2 | 41.9 | No *Logical** | 24.7 | 39.2 | 36.2 |
| 0 | 0.0 | 17.0 | 13.6 | Simple Aggregation | 40.2 | 66.9 | 61.5 |

* Training was unstable and experienced gradient explosion.

As we can see from the table, LAMBDANET achieves significantly better results compared to Deep-Typer. In particular, LAMBDANET outperforms DeepTyper by $14.1\%$ (absolute) for declaration-level accuracy and by $9.6\%$ for occurrence-level accuracy.

Note that the accuracy we report for DeepTyper ($67.4\%$) is not directly comparable to the original accuracy reported in Hellendoorn et al. (2018) ($56.9\%$) for the following reasons. While we perform static analysis and have a strict distinction of library vs. user-defined types and only evaluate both tools on library type annotations in this experiment, their implementation treat types as tokens and does not have this distinctions. Hence, their model also considers a much larger prediction space consisting of many user-defined types—most of which are never used outside of the project in which they are defined—and is also evaluated on a different set of annotations than ours.

## 5.2 PREDICTING USER-DEFINED TYPES

As mentioned earlier, our approach differs from prior work in that it is capable of predicting user-defined types; thus, in our second experiment, we extend LAMBDANET's prediction space to also include user-defined types. However, since such types are not in the prediction space of prior work (Hellendoorn et al., 2018), we implemented two simpler baselines that can be used to calibrate our model's performance. Our first baseline is the type inference performed by the TypeScript compiler, which is sound but incomplete (i.e., if it infers a type, it is guaranteed to be correct, but it infers type `any` for most variables).[6] Our second baseline, called SIMILARNAME, is inspired by the similarity between variable names and their corresponding types; it predicts the type of each variable $v$ to be the type whose name shares the most number of common word tokens with $v$.

The results of this experiment are shown in Table 2, which shows the top-1 and top-5 accuracy for both user-defined and library types individually as well as overall accuracy. In terms of overall prediction accuracy, LAMBDANET achieves $64.2\%$ for top-1 and $84.5\%$ for top-5, significantly outperforming both baselines. Our results suggest that our fusion of logical and contextual information to predict types is far more effective than rule-based incorporation of these in isolation.

## 5.3 ABLATION STUDY

Table 3 shows the results of an ablation study in which (a) we vary the number of message-passing iterations (left) and (b) disable various features of our architecture design (right). As we can see from the left table, accuracy continues to improve as we increase the number of message passing iterations as high as 6; this gain indicates that our network learns to perform inference over long distances. The right table shows the impact of several of our design choices on the overall result.

---

[6]For inferring types from the TypeScript compiler, we use the code provided by Hellendoorn et al. (2018). We found this method had a slightly lower accuracy than reported in their work.

For example, if we do not use *Contextual* edges (resp. *Logical* edges), overall accuracy drops by 14.5% (resp. 25.8%). These drops indicate that both kinds of predicates are crucial for achieving good accuracy. We also see that the attention layer for NPAIR makes a significant difference for both library and user-defined types. Finally, Simple Aggregation is a variant of LAMBDANET that uses a simpler aggregation operation which replaces the attention-based weighed sum in Eq 1 with a simple average. As indicated by the last row of Table 3 (right), attention-based aggregation makes a substantial difference for user-defined types.

## 5.4 COMPARISON WITH JSNICE

Since JSNice (Raychev et al., 2015) cannot properly handle class definitions and user-defined types, for a meaningful comparison, we compared both tools' performance on top-level functions randomly sampled from our test set. We filtered out functions whose parameters are not library types and manually ensured that all all the dependency definitions are also included. In this way, we constructed a small benchmark suite consisting of 41 functions. Among the 107 function parameter and return type annotations, LAMBDANET correctly predicted 77 of them, while JSNice only got 48 of them right. These results suggest that LAMBDANET outperforms JSNice, even when evaluated only on the places where JSNice is applicable.

## 6 RELATED WORK

**Type Inference using Statistical Methods.** There are several previous works on predicting likely type annotations for dynamically typed languages: Raychev et al. (2015) and Xu et al. (2016) use structured inference models for Javascript and Python, but their approaches do not take advantage of deep learning and are limited to a very restricted prediction space. Hellendoorn et al. (2018) and Jangda & Anand (2019) model programs as sequences and AST trees and apply deep learning models (RRNs and Tree-RNNs) for TypeScript and Python programs. Malik et al. (2019) make use of a different source of information and take documentation strings as part of their input. However, all these previous works are limited to predicting types from a fixed vocabulary.

**Graph Embedding of Programs.** Allamanis et al. (2017) are the first to use GNNs to obtain deep embedding of programs, but they focus on predicting variable names and misuses for $C^\sharp$ and rely on static type information to construct the program graph. Wang et al. (2017) use GNNs to encode mathematical formulas for premise selection in automated theorem proving. The way we encode types has some similarity to how they encode quantified formulas, but while their focus is on higher-order formulas, our problem requires encoding object types. Veličković et al. (2018) are the first to use an attention mechanism in GNNs. While they use attention to compute node embeddings from messages, we use attention to compute certain messages from node embeddings.

**Predicting from an Open Vocabulary.** Predicting unseen labels at test time poses a challenge for traditional machine learning methods. For computer vision applications, solutions might involve looking at object attributes (Farhadi et al., 2017) or label similarity Wang et al. (2018); for natural language, similar techniques are applied to generalize across semantic properties of utterances (Dauphin et al., 2013), entities (Eshel et al., 2017), or labels (Ren et al., 2016). Formally, most of these approaches compare an embedding of an input to some embedding of the label; what makes our approach a pointer network (Vinyals et al., 2015) is that our type encodings are derived during the forward pass on the input, similar to unknown words for machine translation (Gulcehre et al., 2016).

## 7 CONCLUSIONS

We have presented LAMBDANET, a neural architecture for type inference that combines the strength of explicit program analysis with graph neural networks. LAMBDANET not only outperforms other state-of-the-art tools when predicting library types, but can also effectively predict user-defined types that have not been encountered during training. Our ablation studies demonstrate the usefulness of our proposed logical and contextual hyperedges.

For future work, there are several potential improvements and extensions to our current system. One limitation of our current architecture is the simplified treatment of function types and generic types

(i.e., collapsing them into their non-generic counterparts). Extending the prediction space to also include structured types would allow us to make full use of the rich type systems many modern languages such as TypeScript provide. Another important direction is to enforce hard constraints during inference such that the resulting type assignments are guaranteed to be consistent.

## ACKNOWLEDGMENTS

We would like to thank DeepTyper authors, Vincent J. Hellendoorn, Christian Bird, Earl T. Barr, and Miltiadis Allamanis, for sharing their data set and helping us set up our experimental comparisons. We also thank the ICLR reviewers for their insightful comments and constructive suggestions. Finally, we would also like to thank Shankara Pailoor, Yuepeng Wang, Jocelyn Chen, and other UToPiA group members for their kind support and useful feedback. This project was supported in part by NSF grant CCF-1762299.

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
