# OpenReview forum: "LambdaNet: Probabilistic Type Inference using Graph Neural Networks"
_ICLR.cc/2020/Conference — Accept (Poster)_

### Official Review · AnonReviewer3 · 2019-10-10
**Official Blind Review #3**

**Rating:** 8

**Review:**

This paper presents a GNN-based method for predicting type annotations in JavaScript and TypeScript code. By constructing a "type dependency graph" that encodes the relationships among variables and appropriately modifying GNNs to the hypergraph setting, the authors show that the good performance improvements can be made.

Overall, this paper is well-written, the experiments are quite convincing and the methods are reasonable and interesting. I believe that there are a few things that need to be clarified within the evaluation, but I would argue that this paper should be accepted.

* It is unclear to me what is the scope of the type dependency graph construction. Is it a whole file? Is it a single function/class? A whole project?

* The DeepTyper paper seems to suggest that it predicts type annotations one function at a time. Does the comparison (in 5.1) use the same granularity as LambdaNet? If for example, LambdaNet looks at whole files, then the comparison is not exact. Could you please clarify? Performing the comparison on equal-sized samples would make sense.

* If my reading of DeepTyper is correct, it processes all identifiers as a single unit. In contrast, this work (correctly, in my opinion), breaks the identifiers into "word tokens". This may be an important difference between the two methods. To test this, an ablation where LambdaNet does *not* split the identifiers, would provide a better comparison among the sequential representation of DeepTyper and the type constraint graph of LambdaNet.

* Some comparison with JSNice is missing (Raychev et al. 2015). The DeepTyper paper suggests that the two approaches are somewhat complementary. It would be useful to know how LambdaNet compares to JSNice too.

* There is some literature that suggests that code duplication exists in automatically scraped corpora and that it hurts the evaluation of machine learning models [a,b]. At the very least, the authors should report the percent of duplicates (if any) in their corpus. Another option would be to _not_ evaluate predictions in any duplicated file.

## Secondary question:

* I do not understand why a separate edge is needed for Subtype() and Assign(). Isn't Assign (a,b) == Subype(a,b)?

* The authors correctly exclude the `any` annotations produced by the TypeScript compiler. Do they also exclude any other annotations? For example, functions that do not return a value (i.e. their return value is `void`) would also need to be excluded. What about `object`?

* I would encourage the authors to make the dataset (source code and extracted graphs), and the LambdaNet code public upon acceptance.

## Minor

* Please capitalize GitHub, TypeScript, etc throughout the paper?

* Sec 4: "we first to compute" -> "we first compute"



[a] Lopes, Cristina V., et al. "DéjàVu: a map of code duplicates on GitHub." Proceedings of the ACM on Programming Languages 1.OOPSLA (2017): 84.
[b] Allamanis, Miltiadis. "The Adverse Effects of Code Duplication in Machine Learning Models of Code." arXiv preprint arXiv:1812.06469 (2018).

**Experience Assessment:**

I have published in this field for several years.

**Review Assessment: Checking Correctness Of Derivations And Theory:**

N/A

**Review Assessment: Checking Correctness Of Experiments:**

I carefully checked the experiments.

**Review Assessment: Thoroughness In Paper Reading:**

I read the paper thoroughly.

---

> ### Author Response · Authors · 2019-11-07
> **Responses to Reviewer 3**
>
> Thank you very much for your comments and questions about our paper! Please see our responses below:
>
> =======
> “It is unclear to me what is the scope of the type dependency graph construction. Is it a whole file? Is it a single function/class? A whole project?”
>
> Response: Our approach takes an entire Typescript project as its input (see section 2.1 paragraph 1). We can make this clearer in the evaluation sections.
>
> =======
> “The DeepTyper paper seems to suggest that it predicts type annotations one function at a time. Does the comparison (in 5.1) use the same granularity as LambdaNet? If for example, LambdaNet looks at whole files, then the comparison is not exact. Could you please clarify? Performing the comparison on equal-sized samples would make sense.”
>
> Response: The original DeepTyper architecture takes a Typescript source file as its input and performs inter-file batching to reduce training and inference time, so we did the same in our evaluation. To compare on equal-sized inputs, we could theoretically concatenate all source files into one large file and run DeepTyper on top of it. However, this is extremely unlikely to make any difference due to the limited memory of the RNN.
>
> =======
> “If my reading of DeepTyper is correct, it processes all identifiers as a single unit. In contrast, this work (correctly, in my opinion), breaks the identifiers into "word tokens". This may be an important difference between the two methods. To test this, an ablation where LambdaNet does *not* split the identifiers, would provide a better comparison among the sequential representation of DeepTyper and the type constraint graph of LambdaNet.”
>
> Response: In our comparison, we made sure both DeepTyper and LambdaNet are using the same naming feature that splits words into tokens, through a modification to DeepTyper’s architecture. We did this because we believe word tokens are clearly better, and our results demonstrate LambdaNet’s advantage over DeepTyper even when DeepTyper is using better naming features.
>
> =======
> “Some comparison with JSNice is missing (Raychev et al. 2015). The DeepTyper paper suggests that the two approaches are somewhat complementary. It would be useful to know how LambdaNet compares to JSNice too.”
>
> Response:  We found it very hard to do a comprehensive comparison with JSNice because their tool predicts over a different and very restricted set of types (mostly primitive Javascript types like number, string, etc), so we did not compare with it. DeepTyper’s original experiment only manually compares with JSNice on 30 functions. We could perform a similar manual comparison against JSNice on randomly selected functions and include this comparison in our revision.
>
> =======
> “There is some literature that suggests that code duplication exists in automatically scraped corpora and that it hurts the evaluation of machine learning models [a,b]. At the very least, the authors should report the percent of duplicates (if any) in their corpus. Another option would be to _not_ evaluate predictions in any duplicated file.”
>
> Response: We investigated this question by running jscpd (a popular code duplication detection tool, https://github.com/kucherenko/jscpd ) on our entire data set and found that only 2.7% code is duplicated. Furthermore, most of these duplicates are intra-project. Thus, we believe that code duplication is not a severe problem in our dataset, and we will include more details about this result in any future revision.
>
> =======
> “I do not understand why a separate edge is needed for Subtype() and Assign(). Isn't Assign (a,b) == Subype(a,b)?”
>
> Response: It is true that Assign is a special case of Subtype from a typing perspective. We differentiate them because these edges appear in different contexts in the graph and, thus, having uncoupled parameters for these two edge types is likely to be beneficial. Moreover, assignments constitute a substantial portion of subtyping constraints, so we have enough data to train an additional edge type.
>
> =======
> “The authors correctly exclude the `any` annotations produced by the TypeScript compiler. Do they also exclude any other annotations? For example, functions that do not return a value (i.e. their return value is `void`) would also need to be excluded. What about `object`?”
>
> Response: As we mentioned in the paper, we only evaluate on type annotations that the programmers have manually added. This means that types like “void” would only be evaluated if the programmer feels it is necessary to add them (which is quite rare but possible).
>
> =======
> “I would encourage the authors to make the dataset (source code and extracted graphs), and the LambdaNet code public upon acceptance.”
>
> Response: Yes, we will.

---

> > ### Comment · AnonReviewer3 · 2019-11-08
> > **Response**
> >
> > Thanks for your prompt response! Your answers make sense and are reasonable.
> >
> > Given those, I would love to see the following experiments/information and given these, I'll change my "Weak Accept" to "Accept".
> >
> > * A rudimentary comparison with JsNice: As someone with an interest in this area, I still don't know which method is state-of-the-art. I would imagine that the JsNice authors would insist that their method is the best, whereas you would insist that this is LambdaNet. Doing a small test (say 50-100 annotations) on JsNice (similar to DeepTyper) would be good enough for me.
> >
> > * Given than you don't filter human annotations, I would like to see a break-down of LambdaNet's performance on (a) Any/void (b) primitive types (c) user-defined types. I believe that this will be useful to understand the extent to which the attention mechanism helps (especially with [c]) and help us disregard Any/void annotations that are deterministically predictable. If possible, comparing these with DeepType's results would be nice (but I don't find it crucial).
> >
> > Hopefully, the above seem reasonable to you too.
> >
> > Other comments:
> >
> > * Your type dependence graph is similar to the CRF graph constructed by JsNice, it might be easy enough to convert your graphs to the feature format of Nice2Predict and clearly establish LambdaNet as the state-of-the-art. I understand that this is non-trivial work, that cannot happen during ICLR's discussion period, but I believe that this would be extremely useful to the PL/SE/ML communities and allow to establish an uncontested state-of-the-art.
> >
> > * Duplicates: (1) One of the main findings of the DejaVu work is that there are many duplicates that are not exactly copy-paste (but small perturbations of other code). The jscpd seems to be looking for exact matches (over tokens). I would still encourage you to run one of the tools of [a] or [b] that check for "softer" duplicates. (2) please, do report the percent of duplicates in the paper in any case.

---

> > > ### Author Response · Authors · 2019-11-12
> > > **New experimental results and responses**
> > >
> > > Thanks for suggesting the additional experiments; these are indeed useful information to include in the future version of our paper. We have run the suggested experiments and list the results below:
> > >
> > > =======
> > > “A rudimentary comparison with JsNice: As someone with an interest in this area, I still don't know which method is state-of-the-art. I would imagine that the JsNice authors would insist that their method is the best, whereas you would insist that this is LambdaNet. Doing a small test (say 50-100 annotations) on JsNice (similar to DeepTyper) would be good enough for me.”
> > >
> > > Results:
> > > Since JsNice cannot properly handle class definitions and user-defined types, for a meaningful comparison, we compared both tools’ performance on top-level functions randomly sampled from our test set. We filtered out functions whose parameters are not library types and manually ensured that all all the dependency definitions are also included. In this way, we constructed a small benchmark suit consisting of 41 functions. Among the 107 function parameter and return type annotations, LambdaNet correctly predicted 77 of them, while JsNice only got 48 of them right. This rudimentary results suggest that our tool is better than JSNice, even when evaluated only on the places where JsNice is applicable.
> > >
> > > =======
> > > “Given than you don't filter human annotations, I would like to see a break-down of LambdaNet's performance on (a) Any/void (b) primitive types (c) user-defined types. I believe that this will be useful to understand the extent to which the attention mechanism helps (especially with [c]) and help us disregard Any/void annotations that are deterministically predictable. If possible, comparing these with DeepType's results would be nice (but I don't find it crucial)”
> > >
> > > Results:
> > > The table below shows a breakdown of LambdaNet’s performance on the top-10 most frequent library types in our test set (the remaining types are omitted). As you can see from this graph, void only consists of a small fraction (< 9%) of all library type annotations, hence it has little effect on the overall library type accuracy. Also, as we mentioned in the evaluation section, we also ignore all ‘any’ annotations when computing the accuracies. We can add a detailed accuracy breakdown graph in our future revision.
> > >
> > > $$
> > >  \begin{array}{|l r r c|}
> > >
> > >  \textbf{Type} & \textbf{Total} & \textbf{Correct} & \textbf{Percent} \\
> > >
> > >  String & 2027 & 1753 & 86.5 \\
> > >
> > >  Array & 1111 & 754 & 67.9 \\
> > >
> > >  Number & 932 & 394 & 42.3 \\
> > >
> > >  Void & 663 & 600 & 90.5 \\
> > >
> > >  Boolean & 599 & 508 & 84.8 \\
> > >
> > >  Function & 501 & 266 & 53.0 \\
> > >
> > >  Promise & 326  & 210  & 64.4 \\
> > >
> > >  Object & 304  & 162  & 53.3 \\
> > >
> > >  Date & 54  & 21 & 38.9\\
> > >
> > >  Map & 54  & 13 & 24.1 \\
> > > \end{array}
> > > $$
> > >
> > > =======
> > > Your type dependence graph is similar to the CRF graph constructed by JsNice, it might be easy enough to convert your graphs to the feature format of Nice2Predict and clearly establish LambdaNet as the state-of-the-art. I understand that this is non-trivial work, that cannot happen during ICLR's discussion period, but I believe that this would be extremely useful to the PL/SE/ML communities and allow to establish an uncontested state-of-the-art.
> > >
> > > Response: Thanks for the suggestion -- we will definitely consider this as an avenue for future work.
> > >
> > > =======
> > > Duplicates: (1) One of the main findings of the DejaVu work is that there are many duplicates that are not exactly copy-paste (but small perturbations of other code). The jscpd seems to be looking for exact matches (over tokens). I would still encourage you to run one of the tools of [a] or [b] that check for "softer" duplicates. (2) please, do report the percent of duplicates in the paper in any case.
> > >
> > > Response: Both tools currently seems not to have support for Typescript. But we will try to do this once such tools are available.

---

> > > > ### Comment · AnonReviewer3 · 2019-11-12
> > > > **Thanks**
> > > >
> > > > Thanks a lot for the prompt update! I've changed my recommendation to "Accept" as promised. I'd love to see these results somewhere in the final version of the paper.

---

### Official Review · AnonReviewer2 · 2019-10-21
**Official Blind Review #2**

**Rating:** 8

**Review:**

= Summary
A method to predict likely type of program variables in TypeScript is presented. It consists of a translation of a program's type constraints and defined objects into a (hyper)graph, and a specialised neural message passing architecture to learn from the generated graphs. Experiments show that the method substantially outperforms sound typing in the TypeScript compiler, as well as a recent method based on deep neural networks.

= Strong/Weak Points
+ The graph representation of the problem is novel, and draws both on core ideas from Hindley-Milner typing (in the subtyping/assignment graph bits) as well as neural ideas (in name similiarity)
+ The neural message passing architecture is adapted to the problem, handling features not present in the standard GNN literature (hyperedges, ...)
+ Experiments compare with relevant baselines and consider interesting ablations, studying the effect of the GNN extensions in detail.
- The hyperparameter selection regime (and the experiments used to find them) is not described

= Recommendation
This is an application-driven paper with nice practical results. The fact that standard neural architectures are extended and adapted to the task, and the way domain knowledge is used to design the graph representation makes this interesting even to people outside the task-specific audience, and hence I strongly recommend acceptance.

= Minor Comments
- page 2: "network's type to be class" -> "to be a class"
- Evaluation Datasets: Did you take duplication in the crawled datasets into account? (Lopes et al. 2017 (DéjàVu: a map of code duplicates on GitHub) suggests that this is particularly problematic for JavaScript/TypeScript)


**Experience Assessment:**

I have published in this field for several years.

**Review Assessment: Checking Correctness Of Derivations And Theory:**

N/A

**Review Assessment: Checking Correctness Of Experiments:**

I carefully checked the experiments.

**Review Assessment: Thoroughness In Paper Reading:**

I read the paper thoroughly.

---

> ### Author Response · Authors · 2019-11-07
> **Responses to Reviewer 2**
>
> Thank you very much for your review! Please see our responses below regarding your comments:
>
> “Evaluation Datasets: Did you take duplication in the crawled datasets into account? (Lopes et al. 2017 (DéjàVu: a map of code duplicates on GitHub) suggests that this is particularly problematic for JavaScript/TypeScript)”
>
> Response: We investigated this question by running jscpd (a popular code duplication detection tool, https://github.com/kucherenko/jscpd ) on our entire data set and found that only 2.7% code is duplicated. Furthermore, most of these duplicates are intra-project. Thus, we believe that code duplication is not a severe problem in our dataset, and we will include more details about this result in any future revision.
>
> ========
> “The hyperparameter selection regime (and the experiments used to find them) is not described”
>
> Response: We selected hyperparameters in a standard way by tuning on a validation set as we were developing our model. We’ll include more details about hyperparameters and hyperparameter selection in any future revision.

---

### Official Review · AnonReviewer1 · 2019-10-23
**Official Blind Review #1**

**Rating:** 6

**Review:**

This paper proposed to use Graph Neural Networks (GNN) to do type inference for dynamically typed languages. The key technique is to construct a type dependency graph and infer the type on top of it. The type dependency graph contains edges specifying hard constraints derived from the static analysis, as well as soft relationships specified by humans. Experiments on type predictions for TypeScript have shown better performance than the previous methods, with or without user specified types.

Overall this paper tackles a nice application of GNN, which is the type prediction problem that utilizes structural information of the code. Also the proposed type dependency graph seems interesting to me. Also the pointer mechanism used for predicting user specified types is a good strategy that advances the previous method. However, I have several concerns below:

About formulation:
1) I’m not sure if the predicted types for individual variable would be very helpful in general. Since the work only cares about individual predictions while no global consistency is enforced, it is somewhat limited. For example, in order to (partially) compile a program, does it require all the variable types to be correct in that part? If so, then the predicted types here might not be that helpful. I’m not sure about this, so any discussion would be appreciated.


About type dependency graph:
1) Comparing to previous work (e.g, Allamanis et.al, ICLR 18), it seems the construction of the task specific graph is the major contribution, where the novelty is a bit limited.
2) The construction of the dependency graph is heuristic. For example, why the three contextual constraints are good? Would there be other good ones? Also why only include such limited set of logical constraints. For example, would expression like (x + y) induce some interesting relationships? Because such hand-crafted graph is lossy (unlike raw source code), all the questions here lead to the concern of such design choices.
3) The usage of graph is somewhat straightforward to me. For example, although the hard-constraints are there, there’s no such constraints reflected in the prediction. Adding the constraints on the predictions would be more interesting.

About experiments:
1) I think one ablation study I’m most interested in is to simply run GNN on the AST (or simply use Allamanis et.al’s method). This is to verify and support the usage of proposed type dependency graph.
2) As the authors claimed in Introduction, ‘plenty of training data is available’. However in experiment only 300 projects are involved. Also it seems that these are not fully annotated, and the ‘forward type inference functionality from TypeScript’ is required to obtain labels. It would be good to explain such discrepancy.
3) Continue with 2), as the experiment results shown in Table 2, TS compiler performs poorly. So how would it be possible to train with poor annotations, while generalize much better? Some explanations would be helpful here.
4) I think only predicting non-polymorphic types is another limitation. Would it be possible to predict structured types? like nested list, or function types with arguments?


**Experience Assessment:**

I have published one or two papers in this area.

**Review Assessment: Checking Correctness Of Derivations And Theory:**

N/A

**Review Assessment: Checking Correctness Of Experiments:**

I assessed the sensibility of the experiments.

**Review Assessment: Thoroughness In Paper Reading:**

I read the paper thoroughly.

---

> ### Author Response · Authors · 2019-11-07
> **Responses about formulation and type dependency graph**
>
> Thank you very much for your comments and questions about our paper! Due to length limit, we break our responses into two comments. Please see the parts about formulation and type dependency graph below:
>
> =======
> “I’m not sure if the predicted types for individual variable would be very helpful in general. Since the work only cares about individual predictions while no global consistency is enforced, it is somewhat limited.”
>
> Response: We believe that predicted types would still be helpful because the user only needs to focus on parts of the codebase that do not type check (combined with techniques for localizing type checking errors). That said, we are definitely interested in using structured prediction techniques (e.g., structured SVM) to enforce that the predicted types pass the type checker at learning and inference time. However, this aspect is orthogonal to the techniques presented in our submission and brings several new research challenges.
>
> =======
> “For example, in order to (partially) compile a program, does it require all the variable types to be correct in that part? If so, then the predicted types here might not be that helpful. I’m not sure about this, so any discussion would be appreciated.”
>
> Response: If the program does not type check, the TypeScript compiler will report a warning but still emit the Javascript program anyway. Thus, mistakes in type error prediction neither affect the compilation process nor cause any additional run-time errors.
>
> =======
> "Comparing to previous work (e.g, Allamanis et.al, ICLR 18), it seems the construction of the task specific graph is the major contribution, where the novelty is a bit limited."
>
> Response: Our architecture is only related to Allamanis et al. insofar as it is a graph neural network over source code. Most components of our model introduce substantial new elements. First, the type embedding scheme for user-defined types is entirely new, which we believe would also be useful for other tasks involving types (e.g. program language models, code repair, static analysis...). Second, the lack of static type information makes the problem fundamentally different and requires much more sophisticated attention mechanism (in the form of usage hyperedges) to help propagating information between syntactically distant but semantically relevant program elements.
>
> =======
> “The construction of the dependency graph is heuristic. For example, why the three contextual constraints are good? Would there be other good ones? Also why only include such limited set of logical constraints. For example, would expression like (x + y) induce some interesting relationships? Because such hand-crafted graph is lossy (unlike raw source code), all the questions here lead to the concern of such design choices. ”
>
> Response: The set of logical constraints contains all the hard constraints that a rule-based type inference algorithm would use. But as our ablation studies show, adding extra statistical information via contextual edges can be very helpful. Our choices of contextual edges aim to preserve the statistical hints that are most relevant to make correct type prediction while exclude type-irrelevant aspects of code that are more likely leading to overfitting. In the case of the expression “x+y”, it would be translated into the Call relation “$\textrm{Call}(N_z,N_+,N_x,N_y)$” asserting that $N_z=N_+(N_x,N_y)$, where $N_x$, $N_y$, and $N_+$ are the nodes corresponding to “x”, “y”, and the library function “+”. And $N_z$ is the node generated for the entire expression. Since $N_+$ is a library node whose embedding vector is fine-tuned during the training process, information can propagate from it into other three nodes via the Call edge and hint the GNN that they are all likely to be of type “number”.

---

> ### Author Response · Authors · 2019-11-07
> **(Continued) Responses about experiments**
>
> “I think one ablation study I’m most interested in is to simply run GNN on the AST (or simply use Allamanis et.al’s method). This is to verify and support the usage of proposed type dependency graph.”
>
> Response: It would be infeasible to apply Allamanis’ architecture in our setting since they make heavy use of type information when constructing the semantic edges. (e.g., if they see “a.f(x)”, since they know the exact type of “a”, they know which method “f” “a.f” is referring to, hence they can directly connect the node for the parameter of the method “f” to the node for the argument “x”). We did not implement a GNN architecture directly runnable on program ASTs, but we did try removing part or all of our semantic edges in our ablation studies, which is as close as we can feasibly get to a direct comparison.
>
> =======
> “As the authors claimed in Introduction, ‘plenty of training data is available’. However in experiment only 300 projects are involved. Also it seems that these are not fully annotated, and the ‘forward type inference functionality from TypeScript’ is required to obtain labels. It would be good to explain such discrepancy.”
>
> Response: We note that 300 projects already contain 1.2 million lines of Typescript code and hundreds of thousands of labeled type variables to predict. Empirically, we found that the performance of our model was already saturating here; with orders of magnitude more data and a bigger architecture, higher accuracy could possibly be achieved, but this would require significantly more computational resources.
>
> =======
> “as the experiment results shown in Table 2, TS compiler performs poorly. So how would it be possible to train with poor annotations, while generalize much better? Some explanations would be helpful here.”
>
> Response: Like in many modern programming languages with forward type inference (e.g., Scala, C#, Swift...), a Typescript programmer does not need to annotate every definition in order to fully specify the types of a program. Instead, they only need to annotate some “key places” (e.g., function parameters and return types, class members) and let the forward inference algorithm to figure out the rest of the types. Therefore, in our training set, we can keep the user annotations on these key places and run the TS compiler to recover these implicitly specified types as additional labels. This is not strictly necessary but can help us get more training signal and accelerate the convergence of training. In our evaluation, the TS compiler no longer has access to the user annotations on those key places (because the goal here is to predict them), hence its forward inference algorithm performs poorly and can only predict ‘any’ for most places.
>
> =======
> “I think only predicting non-polymorphic types is another limitation. Would it be possible to predict structured types? like nested list, or function types with arguments?”
>
> Response: It is possible to modify our architecture to predict structured types like (List<Int>) by making a sequence of predictions over type production rules instead of concrete types. However, in general, predicting polymorphic types also involves dealing with quantified types like $\forall T.\ T \rightarrow T$ (the type signature of the identity function) and bounded polymorphism like $\forall T <: \textrm{SomeType}.\ T \rightarrow \textrm{String}$. Properly encoding and predicting such types is an open research problem in its own right, so we leave this extension to future work.

---

> > ### Comment · AnonReviewer1 · 2019-11-14
> > **Re: Official Blind Review #1**
> >
> > Thank you very much for your detailed responses. I found most of my concerns have been addressed.
> > However I do suggest to put some of the explanations into the paper if applicable. For example, how the TS compiler be useful during training with the user annotation, and why itself performs poorly along.
> >
> > Regarding the technical side, I personally feel the usage of type constraints as extra edges to GNN is not super novel or fundamental, as the GNN can still violate the constraints; probably the adjustments on the modeling side would be more effective, and more data efficient hopefully.
> >
> > Nevertheless, I think the experimental results are solid and the paper has made non-trivial contribution to the literature, so I'm happy to increase my score to 6.

---

### Decision · Program_Chairs · 2019-12-19

**Decision:**

Accept (Poster)

**Comment:**

This paper proposes an approach to type inference in dynamically typed languages using graph neural networks. The reviewers (and the area chair) love this novel and useful application of GNNs to a practical problem, the presentation, the results. Clear accept.